# "It just seemed like a perfect storm": A multi-methods feasibility study on the use of Facebook, Google Ads, and Reddit to collect data on abortion-seeking experiences from people who considered but did not obtain abortion care in the United States

Heidi Moseson[1]*, Jane W. Seymour[2¤], Carmela Zuniga[2], Alexandra Wollum[1], Anna Katz[1], Terri-Ann Thompson[2], Caitlin Gerdts[1]

1 Ibis Reproductive Health, Oakland, California, United States of America, 2 Ibis Reproductive Health, Cambridge, Massachusetts, United States of America

¤ Current address: University of Wisconsin, Madison, Wisconsin, United States of America
* hmoseson@ibisreproductivehealth.org

## Abstract

Most studies of abortion access have recruited participants from abortion clinics, thereby missing people for whom barriers to care were insurmountable. Consequently, research may underestimate the nature and scope of barriers that exist. We aimed to recruit participants who had considered, but failed to obtain, an abortion using three online platforms, and to evaluate the feasibility of collecting data on their abortion-seeking experiences in a multi-modal online study. In 2018, we recruited participants for this feasibility study from Facebook, Google Ads, and Reddit for an online survey about experiences seeking abortion care in the United States; we additionally conducted in-depth interviews among a subset of survey participants. We completed descriptive analyses of survey data, and thematic analyses of interview data. Recruitment results have been previously published. For the primary outcomes of this analysis, over one month, we succeeded in capturing data on abortion-seeking experiences from 66 individuals who were not currently pregnant and reported not having obtained an abortion, nor visited an abortion facility, despite feeling that abortion could have been the best option for a recent pregnancy. A subset of survey respondents (n = 14) completed in-depth interviews. Results highlighted multiple, reinforcing barriers to abortion care, including legal restrictions such as gestational limits and waiting periods that exacerbated financial and other burdens, logistical and informational barriers, as well as barriers to abortion care less frequently reported in the literature, such as a preference for medication abortion. These findings support the use of online recruitment to identify and survey an understudied population about their abortion-seeking experiences. Further, findings contribute to a more complete understanding of the full range of barriers to abortion care that people experience in the United States, and how these barriers intersect to not just delay, but to prevent people from obtaining abortion.

**Data Availability Statement:** Data cannot be shared publicly because of identifiability concerns given nature of questions asked, and confidentiality promises made during the informed consent process. De-identified and pared down data can be made available to researchers who meet the criteria for access to confidential data and provide a reasonable request. Interested researchers, can request access to the data via addition to our IRB protocol by contacting our data access committee, chaired by Allie Wollum (awollum@ibisreproductivehealth.org), and by contacting the IRB for the study: the Allendale Investigational Review Board by phone at +1-860-434-5872 or via email (Rta1ali1@aol.com).

**Funding:** HM, CG received a grant from the Society of Family Planning (grant#: SFPRF11-11) to fund this work (https://societyfp.org/). The funders had no role in study design, data collection and analysis, decision to publish, or preparation of the manuscript.

**Competing interests:** The authors have declared that no competing interests exist.

## Introduction

In the United States, people have difficulty accessing abortion care because of a range of barriers, including cost of services and ancillary expenses; gestational age restrictions; legal restrictions; long distances to care; and lack of abortion-related information [1–12]. Although these barriers have negative consequences individually [5], people seeking abortion often experience them in combination, which compounds their effect [13]. These overlapping barriers result in higher costs, and thus may be particularly burdensome to the majority of US abortion patients who are low-income or live below the Federal Poverty Level [12].

Although many studies have examined barriers to abortion, most have recruited participants from abortion-providing facilities, thereby excluding people for whom barriers to care were insurmountable. As a result, existing research may underestimate the extent to which certain factors act as barriers to abortion, and there may be unidentified barriers to care or interactions between barriers. Recognizing the limitation of recruiting only from abortion clinics, a study in Louisiana and Maryland concluded that recruitment in prenatal clinics is a feasible way to find people who considered, but did not obtain, an abortion [14]; approximately one-third of participants considered, but did not obtain an abortion, citing personal preferences and policy-related barriers. However, even the expansion to prenatal clinics may still miss portions of the target population, including people who miscarry, who have not yet entered into care, or who feel they have no source of care or cannot access care without increasing risks of deportation or other privacy concerns. Addressing many of these concerns, a recent study recruited currently pregnant participants searching for information on abortion from Google Ads: at the one-month follow-up, less than half of participants (48%) had had an abortion [15,16].

The harms of not obtaining wanted abortion care are well established [17–22]; yet, without understanding the full range and impact of barriers to abortion care, public health practitioners and policymakers cannot develop appropriate interventions to improve access. With this multi-methods feasibility study, we used three online platforms (Facebook, Google Ads, and Reddit) to recruit a narrow population of people who have been left out of most prior research on abortion access: namely, people who had considered, but not obtained an abortion, nor made it to an abortion clinic, and to assess their experiences with accessing abortion care. We hypothesized that: (1) we would successfully find and recruit members of this population using online platforms, and (2) would be able to capture new information on the number and nature of barriers to abortion care experienced by this understudied population. We have previously published results regarding the first hypothesis: we found that these platforms can indeed identify and recruit the target population [23]. This study reports results related to the second hypothesis: the ability to collect data on barriers to abortion care from this previously excluded population.

## Materials and methods

### Recruitment

Between August 15 and September 15, 2018, we recruited for a brief online survey through advertisements on Facebook, Google Ads, and two Reddit threads (birth control and menstruation; the abortion thread did not allow researchers to post for study recruitment). The one-month recruitment period mirrored a recent abortion-related study that recruited using Google Ads [23]. A digital marketing firm, BUMP Recruitment, managed the posting and purchasing of Facebook and Google Ads advertisements. Study authors managed the Reddit campaigns. Advertisements in English and Spanish (the two most widely spoken languages in

the United States) read: "Complete a 5 minute survey about unplanned pregnancy and be entered to win $50 gift card" and offered a link to complete an "Unplanned Pregnancy Study." The advertisements led to a study website with additional study information and a link to the eligibility screener. If eligible, individuals were offered the opportunity to complete the short online survey. At survey completion, participants could indicate their interest in an in-depth interview. Further details of recruitment methods and results have been previously published [23].

## Survey details and analysis

To recruit the narrowly defined population of interest, the survey screener identified eligible participants who were: aged 15 to 49 years old and English or Spanish speaking US residents. In addition, respondents had to report at least one pregnancy in the past five years for which they felt abortion was the best option, but did not obtain an abortion (nor visit an abortion clinic) for any pregnancy in the past five years. Eligibility questions included: "Did you consider abortion for any of these pregnancies, even for just one second?" and "If it had been available to you, could abortion have potentially been the best option for any of these pregnancies?" An individual needed to respond "yes" to both of these questions to be eligible. The survey was administered via Qualtrics (Qualtrics, Provo, UT) and included up to 28 open- and closed-ended questions about experiences with unwanted pregnancy, considering abortion, obtaining abortion care, and sociodemographic characteristics (S1 File). Adaptive questioning reduced the number of questions based on responses; questions were displayed between three and 12 screens. To minimize fraudulent responses designed to avoid skip logic patterns, respondents could not go back to revise responses. Study team members pre-tested the questionnaire's technical functionality before study launch.

Respondents reported number of pregnancies in the last five years, for how many pregnancies they considered abortion, for how many pregnancies abortion could have been the best option, and barriers they faced when seeking abortion care. In addition to selecting from pre-specified barriers, participants could write-in barriers and were asked, "Please tell us in your own words about why you did not obtain the abortion." Participants who completed the survey were entered into a raffle for a single $50 gift card. For this feasibility study, we aimed to recruit a minimum of 10 participants per recruitment platform, or a minimum of 30 total participants across Facebook, Google Ads, and Reddit in the one-month period.

We conducted descriptive analyses of data from closed-ended survey questions using Stata version 15.1 (Stata, StataCorp, College Station, TX), and summarized open-ended question responses. Where appropriate, closed-ended responses were recoded based on open-ended responses. We report results in accordance with the Checklist for Reporting Results of Internet E-Surveys (CHERRIES) [24].

## In-depth interview details and analysis

All participants who completed the quantitative survey were asked if they were interested in participating in an in-depth interview; if interested, participants provided their name or pseudonym, email address or phone number, and preferred language. The research team contacted all participants who expressed interest to schedule an interview. Five cisgender women who identified as Afrolatina, Asian, white, and/or Latinx, fluent in English (and two also in Spanish), resided in California or Massachusetts, and were trained in in-depth interviewing, conducted all 30–60 minute interviews. Participants who completed the interview received a $25 gift card. As a formative study, we aimed to capture a small sampling of experiences to identify areas for further research, rather than a pre-specified sample size. We conducted an in-depth interview with all interested participants who responded to investigator outreach.

Using a semi-structured guide, interviewers asked participants about: their experiences with pregnancy; circumstances, emotions, and decision-making processes related to their most recent pregnancy for which abortion may have been the best option; factors or barriers that led them to not have an abortion; and, if abortion was considered for more than one pregnancy, any differences in circumstances compared to their most recent pregnancy for which they considered abortion; and sociodemographic characteristics. Interviews were conducted and audio-recorded via a secure online platform, and subsequently transcribed. We analyzed de-identified transcripts thematically, using an iteratively adapted codebook, which initially contained a priori identified codes. To ensure consistency, two transcripts were coded independently by three reviewers, and discrepancies were resolved prior to coding the remaining transcripts. We coded interviews in Dedoose (Dedoose Version 8.1.8 (2018) Los Angeles, CA).

The survey and in-depth interview guide were designed concurrently to complement one another. We analyzed survey data prior to interview data. We note convergence between survey and interview findings in the results. Along with quotes from open-ended survey questions or interviews, we include participants' age, self-reported race or ethnicity, and U.S. Census Region for context.

### Ethical review

To minimize risk of breach of participant confidentiality, all participants gave digital consent to participate in the survey. Similarly, participants gave verbal consent to participate in the interviews; verbal consent was audio-recorded in the audio file, and also documented in a printed consent form, signed and dated by the interviewer. This study, including informed consent processes, was approved by the Allendale Investigational Review Board.

## Results and discussion

### Survey results

**Participant characteristics.** Out of 1,254 eligibility screener submissions, the final analytic sample for these analyses included 66 participants (68.5% of those eligible) from 30 states (Fig 1). Participants were 29 years old on average, the majority identified as female and white, and eight (12%) as Hispanic or Latinx. Nearly two-thirds lived in the Midwest or South Census regions, and 53 (80%) had health insurance (Table 1). Most participants in this sample found the study through Facebook (86%). Nine participants (14%) skipped the final survey section on sociodemographic characteristics: two of these respondents took the survey in Spanish and seven in English.

**Barriers to abortion care.** From the multiple choice list of barriers to abortion, all participants reported at least one listed barrier, and 52% (n = 34) experienced two or more, with 29% (n = 19) reporting three or more. The most commonly reported barriers were ability to pay (56%), concern about judgement (45%), and ability to locate a provider (35%) (Table 2), many of which were experienced simultaneously (Fig 2).

In open-ended responses, 19.7% (n = 13) of participants described additional reasons they did not obtain an abortion. Most (n = 12) noted personal reasons, including 12.7% (n = 8) who chose to continue the pregnancy, 3.0% (n = 2) who would "hate" themselves for having an abortion, 1.5% (n = 1) who "wouldn't be able to handle it," and 1.5% (n = 1) who felt it was wrong to terminate because they were marrying the other person involved in the pregnancy. Additionally, participants described wanting an abortion so they could more readily leave an abusive relationship, or to better care for the children they already had. Others elaborated on the overlapping barriers posed by policy requirements (the additional consultation visit), time

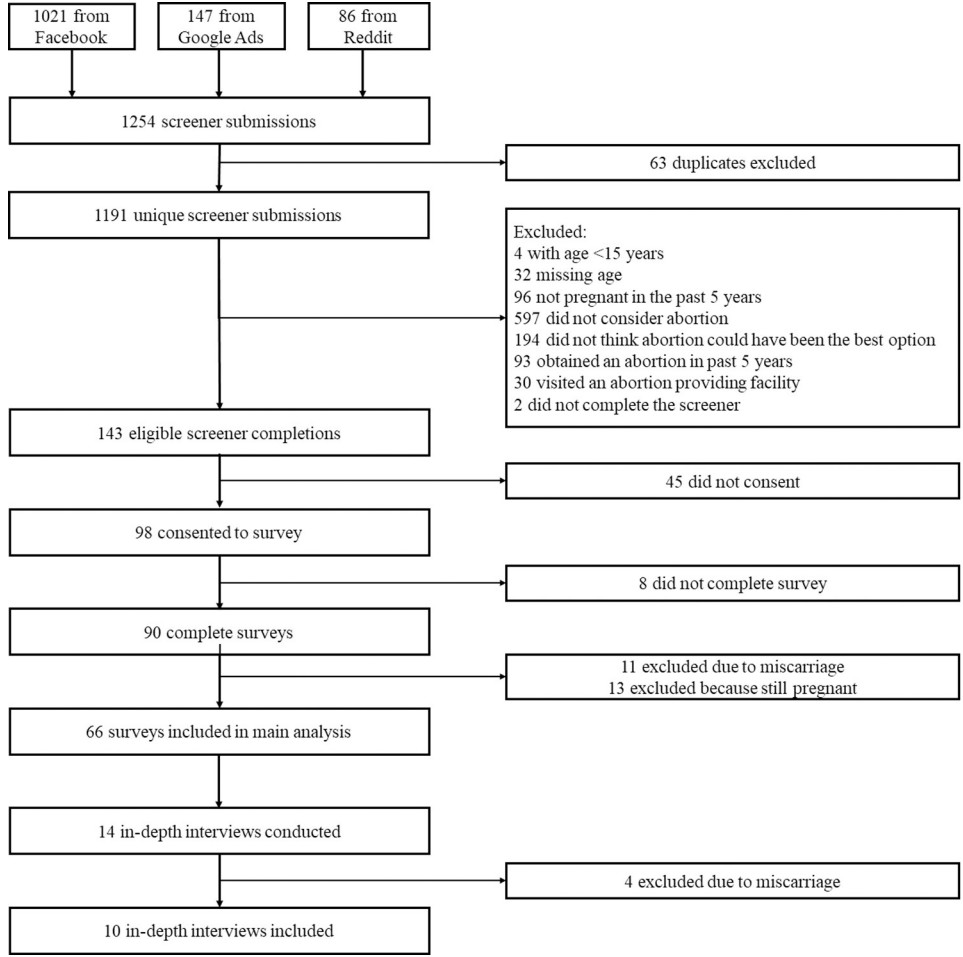

**Fig 1. Recruitment flow diagram.** Participant pathways from screening to survey to interview.

off work, loss of wages, and also stigma. A participant from the South (white, 24 years old) described it: "The nearest provider was in another state. I would've had to have a consultation appointment before an abortion. It would have caused me to take too much time off work and lose too much money. Also many people already knew about the pregnancy and would've judged me." Another respondent from the South (white, 28 years old) described informational barriers to abortion care stating that "I could not find a provider close to me that would answer my questions without an appointment. And research online gave websites that had false information presented like facts that basically scared me into changing my mind." These findings were echoed and elaborated on in the in-depth interviews.

## In-depth interview results

**Participant characteristics.** We conducted 14 interviews. Ten interviews were included for this analysis; we excluded four interviews from participants who reported a miscarriage as the primary reason that they did not obtain abortion. Nine participants disclosed their age at the time of their most recent pregnancy for which they considered abortion; ages ranged from 19 to 43 years (Table 3). Five participants discovered their pregnancy at or before 6 weeks gestation, while the other five discovered their pregnancies at 8, 9, 10, or 12 weeks gestation.

**Table 1. Characteristics of eligible survey participants among an online sample of people who considered, but did not obtain, abortion care for a recent pregnancy (n = 66).**

| Participant Characteristics | *n* | *%* |
|---|---|---|
| *Sex* | | |
| Female | 57 | 86 |
| Missing | 9 | 14 |
| *Age* | | |
| Average, SD | 29 | 6 |
| Minimum, maximum | 19 | 44 |
| Missing | 0 | 0 |
| *Survey language* | | |
| English | 63 | 95 |
| Spanish | 3 | 5 |
| *Recruitment source* | | |
| Facebook | 57 | 86 |
| Google Ads | 2 | 3 |
| Reddit | 6 | 9 |
| Friend shared the link | 1 | 2 |
| *Census region** | | |
| South | 27 | 41 |
| Midwest | 16 | 24 |
| West | 9 | 14 |
| Northeast | 5 | 8 |
| Missing | 9 | 14 |
| *Relationship status* | | |
| Single | 9 | 14 |
| In a relationship | 21 | 32 |
| Married/civil union | 26 | 40 |
| Separated/widowed/divorced | 3 | 5 |
| Missing | 9 | 14 |
| *Education* | | |
| Less than high school graduate | 3 | 5 |
| High school graduate or GED | 11 | 17 |
| Some college or Associates degree | 27 | 41 |
| Bachelor's degree or more | 10 | 15 |
| Missing | 9 | 14 |
| *Race*** | | |
| American Indian or Alaska Native | 4 | 6 |
| Middle Eastern or North African | 1 | 2 |
| Native Hawaiian or Pacific Islander | 1 | 2 |
| White | 49 | 74 |
| Missing | 13 | 20 |
| *Hispanic* | | |
| Yes | 8 | 12 |
| Missing | 9 | 14 |
| *Proportion of the time you had enough money to meet basic living needs last month* | | |
| All/most of the time | 35 | 53 |
| Some of the time/rarely/never | 22 | 33 |
| Missing | 9 | 14 |
| *Current insurance status* | | |
| Public/government | 31 | 47 |

(*Continued*)

**Table 1.** (Continued)

| Participant Characteristics | *n* | % |
|---|---|---|
| Private | 22 | 33 |
| Uninsured | 4 | 6 |
| Missing | 9 | 14 |

* https://www2.census.gov/geo/pdfs/maps-data/maps/reference/us_regdiv.pdf.

** Participants could select more than one response.

**Barriers to abortion care.**   Consistent with survey results, participants described multiple barriers that prevented them from obtaining abortion care. In most cases, two or more barriers exacerbated one another.

*Financial barriers.* All but one participant mentioned the cost of abortion as a barrier to care. Participants described low hourly wages and a lack of savings that precluded their ability to pay for care, partners who did not or would not contribute to abortion costs, and having to decide between paying essential monthly bills and paying for an abortion. Some noted the irony of cost being a barrier, given the low cost of abortion relative to parenting. One participant from the Midwest (white, 17 years old at time of pregnancy) highlighted how lack of insurance coverage for abortion, possibly as a result of policy restrictions on public funding for abortion, made the cost of abortion too high:

> [. . .] insurance did not cover [abortion] at all, so everything would be out-of-pocket. And any complications that arose from it would also not be covered by insurance. It sounds dumb at the time, but it was more expensive for me to go through with an abortion than it was to just keep the baby, which after you have the baby, obviously it's a lot more expensive than just terminating a pregnancy. But it's just one of those things that if you don't have the [money] upfront for it, then they can't really do it.

Participants also described how financial costs layered on top of other barriers, including a lack of information about abortion cost, gestational limits, stigma from partners or family, and a need for time off from work and/or childcare. Most participants did not have accurate information on the cost of abortion, either because clinics would not quote a price over the phone

**Table 2. Barriers to abortion access among an online sample of people who considered, but did not obtain, abortion care for a recent pregnancy (n = 66).**

| | Past pregnancies |
|---|---|
| | *n (%)* |
| Could not pay | 37 (56) |
| Worried about judgment | 30 (45) |
| Could not locate nearby provider | 22 (33) |
| Could not get time off from work | 9 (14) |
| Too far in pregnancy | 6 (9) |
| Personally opposed to abortion | 6 (9) |
| Could not find childcare | 3 (5) |
| Another barrier | 14 (21) |

Note: Columns may not total 100%, as participants could select more than one response.

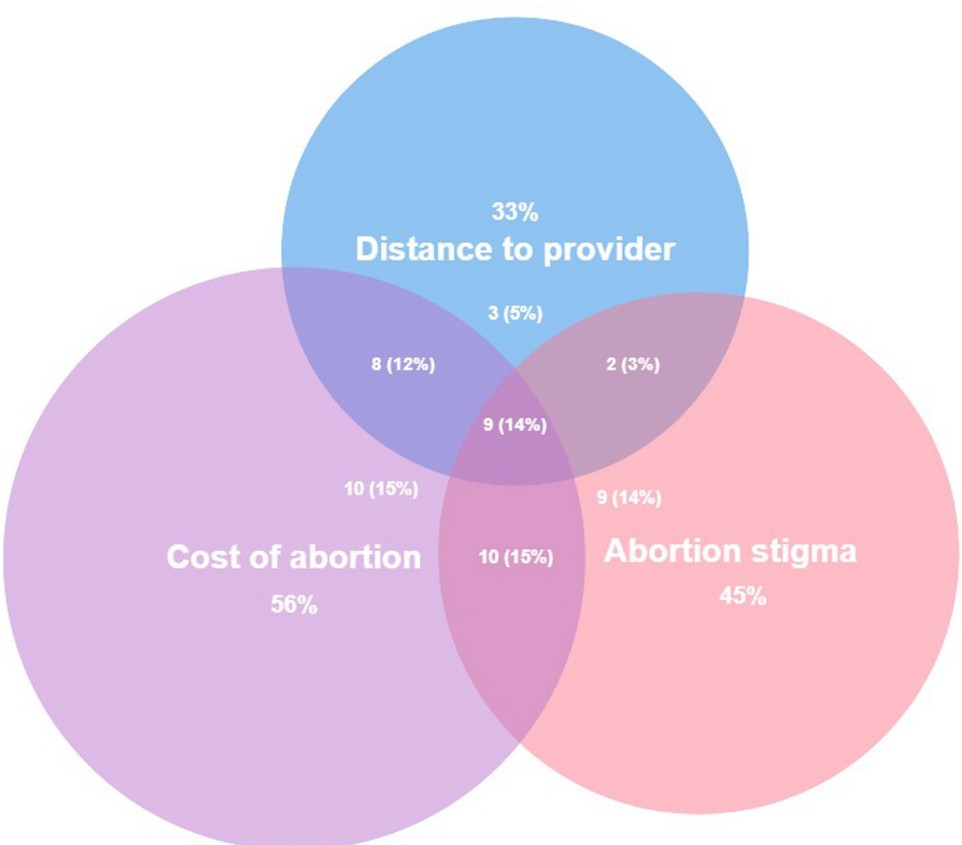

**Fig 2. Overlapping barriers to abortion care.** A Venn diagram depicting overlap across the three most frequently reported barriers to abortion care among an online sample of people who considered, but did not obtain, abortion care for a recent pregnancy.

or they found conflicting information in online forums and Internet search results. While some participants were aware of abortion funds, one did not reach out to a fund due to uncertainty as to how it might work with her insurance. This lack of information further complicated participants' planning. Several described that financial challenges were exacerbated by their understanding of the gestational limit on abortion in their state, meaning that they had a short window of time to gather the needed funds, and restrictions on public funding for abortion.

*Policy-related barriers*. Nearly all participants discussed policy-related barriers to care that affected their abortion-seeking experience; in particular, waiting periods, gestational limits, and restrictions on public funding for abortion. One participant explicitly stated that abortion-related laws acted as a "roadblock" to obtaining abortion, because she could not get information she needed to navigate the conflicting and varied restrictions by state. As a result of gestational limits, some participants felt time pressure to decide whether abortion was right for their pregnancy, believing that they only had a few weeks from a later discovery of the pregnancy until the gestational limit for abortion in their state. In some cases, participants' understanding of the gestational limit in their state was not accurate, but still influenced their decision-making. One participant from the Midwest (Pacific Islander, 25 years old at the time of pregnancy) incorrectly believed she was only days away from the gestational limit, so ended up continuing the pregnancy because she could not make the decision quickly enough:

**Table 3. Characteristics of eligible in-depth interview participants (n = 10), among an online sample of people who considered, but did not obtain, abortion care for a recent pregnancy.**

|  | N |
|---|---|
| *Gender identity* | |
| Cisgender woman | 10 |
| *Age at time of interview, years* | |
| Average | 30 |
| Minimum, maximum | 20, 45 |
| *Census region** | |
| Northeast | 0 |
| South | 4 |
| Midwest | 4 |
| West | 2 |
| *Education* | |
| Less than high school graduate | |
| High school graduate or GED | 2 |
| Some college or Associates degree | 3 |
| Bachelor's degree or more | 5 |
| *Number of children* | |
| 0 | 0 |
| 1 | 4 |
| 2 | 2 |
| 3 | 3 |
| 6 | 1 |
| *Race*** | |
| American Indian or Alaska Native | 1 |
| Native Hawaiian or Other Pacific Islander | 1 |
| White | 8 |
| *Hispanic* | |
| Yes | 3 |
| *Age at first pregnancy* | |
| 17–19 | 4 |
| 20–22 | 4 |
| 23–25 | 2 |
| *Number of pregnancies in the past 5 years* | |
| 1 | 2 |
| 2 | 3 |
| 3 | 3 |
| 4 | 2 |
| *Number of pregnancies in the past 5 years for which abortion could have been the best option* | |
| 1 | 8 |
| 2 | 2 |

* https://www2.census.gov/geo/pdfs/maps-data/maps/reference/us_regdiv.pdf.

** Participants could select more than one response.

*So I just wish that I had a little bit more time to think about it [. . .]–when I was on week 11 of the pregnancy we were really on the fence about it and I just wish I had more time to think about it.*

Another participant said that she and her partner could not obtain enough money for the abortion before the gestational limit. If that policy had not been in place, she felt she could have raised enough money for the abortion; but because she experienced both barriers–cost and the gestational limit–abortion became unobtainable. Another participant described the mandatory counseling and waiting period laws in her state as a deterrent to seeking abortion care. To comply with the required additional clinic visit, this participant would have had to pay for an overnight stay in a hotel, take more time off from work, and cover childcare costs– the waiting period policy thus worsened the financial burden of seeking care.

*Logistical barriers.* Many participants mentioned logistical barriers to abortion, such as an inability to take time off work or requirements to disclose to their employer the reason for time off, an inability to find childcare, or time required for travel. Participants discussed how policy and financial barriers exacerbated the barriers imposed by these logistical factors. For instance, the need for time off work and childcare was often compounded by large distances to the nearest abortion clinic; many reported drive times of 3.5–4 hours to the nearest facility and needing at least two nights in a hotel, which increased the ancillary costs of abortion and made it financially unobtainable. A participant from the South (white, 25 years old at the time of her pregnancy) described multiple such barriers: the logistical barrier of a four hour drive across state lines to the nearest abortion provider; the social barrier of anti-abortion stigma from her partner, sister, and other family members; the logistical barrier of her partner traveling for work for months at a time, leaving her alone with her children; and an additional logistical barrier of the inability to obtain childcare for her children, one of whom had special needs:

> *I still look at my [child] all the time and think like I wish I had–I know this sounds really awful–I just wish I would have tried harder [to have the abortion]. I wish I would have offered someone some money to watch my [other child] or tried to convince someone–tried to plead with my sister or someone to come out to watch him. . . .at the time, I did the best I could do, but . . . it just seemed like a perfect storm of everything coming together so that I couldn't have the procedure done.*

*Informational barriers.* Participants described a lack of information about abortion. The most frequently reported information gaps included not knowing the cost of abortion care, details about the procedure and what to expect, and where or how to find an abortion-providing facility. Other information gaps included knowledge of gestational age eligibility, any long-term health effects of abortion, and whether legal restrictions existed in their state. Many interviewees did not have all of the information about abortion they wanted. The information that participants obtained came from online searches and forums, and the experiences of friends. Even health care providers were not always a trusted or reliable source of information; several participants mentioned frustration with providers who never acknowledged abortion as an option, omitted information about abortion from counseling, or explicitly refused to provide information based on their personal opposition to abortion. One participant from the Midwest (white, 17 years old at the time of her pregnancy) described such an interaction, saying:

> *I mean, my doctor gives me every sort of treatment option for like my depression. She gave me options for hypothyroidism. She's given me options for everything. But whenever it came to pregnancy it was 'Congratulations,' and that was it. . .. They just kind of skated on by and just figured that I was keeping the baby, and that was that.*

Relatedly, two participants received counseling from Crisis Pregnancy Centers (CPCs) without knowing that CPCs did not provide abortions, and reported confusion as to why

abortion was not mentioned. In these cases, social stigma toward abortion and assumptions about pregnancy desires directly influenced participants' ability to access thorough and accurate information about abortion. Notably, two respondents stated that they might have made a different choice about their pregnancy had they known more about the cost of abortion, where they could obtain care, and up to what gestational age abortion was available.

*Stigma barriers.* Participants discussed strong perceived and felt abortion stigma from their partners, families, communities, and health care providers. The desire to avoid judgment and social ostracism delayed participants' decision-making past gestational limits, or deterred them from seeking abortion. Participants described internalized stigma that stemmed from a religious upbringing and was perpetuated by their communities. This interplay of individual and community stigma complicated and lengthened participants' decision-making, and restricted who they could approach for financial and informational support. A participant from the South (Hispanic, 25 at the time of the pregnancy) referenced financial barriers, lack of knowledge, and internalized stigma as barriers that prevented her from obtaining an abortion:

> *And just how do I drum up money for an abortion? Do I pay my rent or do [I] pay for an abortion? Because my understanding of it was that it was [a] rather expensive option. So I guess I feel like I may have been misinformed about the price at the time, but I pretty much just decided based on I guess social stigmas I had placed on myself; that I should have a baby who is now a toddler who is screaming outside of the door.*

*Abortion method preference.* Some respondents stated that medication abortion was the only acceptable method of abortion for them. In particular, three participants described an aversion to surgical abortion. This preference made participants hesitant to seek abortion care; for some, policy- and information-related barriers intersected with abortion method preferences because they were afraid of being beyond the gestational limit for medication abortion, while for others information-related barriers intersected with preferences because of a misconception that a surgical procedure would be painful for the fetus. One participant from the South (Hispanic, 19 years old at the time of her pregnancy) highlighted how policy, information gaps, and method preference barriers interacted, saying:

> *. . .because I didn't want to get an abortion past 12 weeks—or I think is it eight weeks that you can do the pill? I didn't want to go past the pill mark basically, because I don't think I could have gone through further in a pregnancy with the secondary option of abortion. . .So I just knew that for me, I had to make a choice within the time period for the pill—for the abortion pill.*

**Consequences of failure to obtain abortion.** Across interviews, participants described varied ways in which carrying an unplanned pregnancy to term and raising an unexpected child impacted their lives. Some described shifts to education plans and work ability, and others described emotional and social consequences, including for their relationship with the child. One participant from the South (white, 25 years old at the time of pregnancy) described difficulty in bonding with the pregnancy and the baby:

> *"Sometimes it's really hard to talk about because my–the one that I wanted the abortion for, he's 2 and he's running around the house right now. [. . .] I never bonded with him during my pregnancy because I just didn't want him. And it took a lot of–it was really a strain to try to bond with him after he was born. Even when I went into labor with him, I almost had him in*

*the parking lot because I just didn't even want to go in the hospital and do it all. I didn't want to feed him and I didn't want to hold him.[. . .] I just didn't want to do it all over again."*

## Discussion

In this multi-methods feasibility study, we evaluated whether it was possible to collect data on abortion-seeking experiences from an understudied population recruited via three online platforms. In a one-month recruitment period, we succeeded in identifying and enrolling 66 individuals that met these criteria, and in collecting data on the nature and extent of barriers to abortion care that they had faced. Specifically, participants described multiple, intersecting barriers to abortion care which ultimately deterred them from seeking, or prevented them from obtaining, an abortion in the United States. Future abortion access studies should sample not just from those who present at abortion-providing facilities, but from all those who consider abortion to better understand the magnitude and scope of barriers to care, and to identify the full range of possible intervention points for dismantling barriers and increasing access to abortion–a safe and essential public health service [25]. The online recruitment methods utilized in this study can be utilized to conduct this research.

Implementing these underutilized online recruitment approaches, we identified many barriers similar to those identified in prior research involving participants recruited from abortion clinics, such as the inability to pay for the abortion, long distances to the nearest provider, logistical difficulties, legal restrictions on abortion, and abortion stigma [1–10,13]. Similarly, barriers related to inaccurate information about gestational and other legal limits identified in this feasibility study build on prior research that explores information gaps related to finding and accessing abortion care [6,8]. The magnitude of the barriers identified in this feasibility study, however, and the extent of overlap across barriers may be of greater magnitude than previously measured: nearly one-third of participants reported experiencing three or more barriers. However, due to the nature of study questions, it is not possible to disentangle greater or lesser motivation to obtain abortion care from the magnitude of barriers faced.

Additionally, one barrier identified by this study–medication abortion preference–has not been widely emphasized in the existing literature. This preference was so strong as to drive some participants to continue an unwanted pregnancy and parent, rather than have a surgical abortion due to misconceptions about safety, future fertility implications, or fetal pain. Further, study findings highlight the burden of secondary abortion costs, the shortened timeline for abortion fundraising imposed by gestational limits (or individual misconceptions about gestational limits), and the powerful influence of fear of stigma from health care providers, partners, employers or family on deterring abortion seeking.

Of note, despite the advantage of high insurance coverage in the sample, participants experienced multiple barriers that did not just delay, but outright prevented them from obtaining an abortion. This is cause for concern given the established harms of being denied abortion care [17–22], including ramifications for socioeconomic status, achieving aspirational one-year plans, perceived stress, future pregnancies, and risk of violence from the man involved in the pregnancy [17–20].

Targeted interventions could reduce these barriers. To address information gaps, general health care providers–not just those family planning providers–could be trained to discuss pregnancy options, including abortion, for patients at routine annual exams and in contraception counseling, to ensure that people have abortion information before they need it [8]. Provider counseling could include information about abortion cost, gestational limits and other restrictions, and abortion funds to help cover the financial and practical costs of care. Similarly, advocates could focus on creating educational campaigns to spread awareness of

abortion options and laws, as well as abortion funds, to reduce the number of people deterred from abortion-seeking because of a lack of information or funds.

Policy-related interventions could include the expansion of gestational limits for medication abortion, as the scientific evidence confirms that medication abortion is safe at a wider range of gestational ages than for which it is currently approved [26]. This would help ensure that a medication abortion preference is not a barrier to accessing facility-based abortion. Policymakers could also remove waiting period requirements and other laws that result in the need for multiple clinic visits, as these barriers deterred people from seeking care due to the additional monetary and time costs, and fear of stigma from repeated clinic visits. Other policy-related implications include the lack of insurance coverage for abortion, and the resulting financial strain for many participants that exacerbated other barriers.

## Limitations

This feasibility study has limitations. First, the study population is unlikely to be representative of all individuals who considered but did not obtain abortion care. The narrowly defined analytic sample was small, and included no Asian or Black participants. This is in stark contrast to the racial and ethnic composition of abortion patients nationally, among whom approximately 28% are Black, 25% Hispanic, and 6% Asian or Pacific Islander [27]. For a more direct comparison, a recent study that recruited abortion-seekers using Google Ads had greater success in recruiting Black or African-American as well as Asian participants (28.7% and 2.1% of their sample, respectively)–suggesting gaps in our advertisement campaign, rather than the inability of these online methods to recruit a more racially diverse sample [15]. Indeed, the lack of racial and ethnic diversity in our sample may partially have resulted from the set of advertisements most frequently displayed by Facebook, which excluded the advertisement sets with images of Asian and/or Black individuals. Future studies could better control the diversity of advertisements displayed by creating dedicated campaigns for each advertisement with separate, dedicated funds, and prioritizing advertisement posting to groups and pages that center the experiences and interests of people who hold multiple racial and ethnic identities other than "white".

Second, we saw a non-negligible degree of non-consent among eligible respondents. Approximately 31% of eligible screener respondents did not consent to participate; this proportion is similar to a recent study recruiting abortion seekers using Google Ads that saw 25% non-consent [15]. Understanding which respondents do not consent, and why, could add important information for assessing broader generalizability of these findings.

Third, recruitment may have been affected by retrospective reporting of pregnancy wantedness or desire for abortion. We know that retrospective assessment of pregnancy intention changes for those who are denied a wanted abortion in the clinic setting [28]. However, to be eligible for the current study, individuals had to express interest in a study about unplanned pregnancy and then report that abortion may have been the best option for at least one pregnancy. For pregnancies that participants continued and (many) went on to parent a resulting child, this is a high bar. Thus, we may have missed people with the experiences of interest, but who do not describe or recall their experiences in alignment with screening questions. Future studies should examine alternative screening questions to assess how such questions affect the sample, or prospective studies should be implemented.

## Conclusions

Despite the above limitations, this feasibility study succeeded in collecting data from a narrowly defined and understudied population recruited via three online platforms, and the

findings presented here can inform future research among larger samples to ensure greater diversity across participant experiences and identities, and to uncover lesser studied barriers to abortion care and the ways in which barriers interact to reinforce each other. Despite some challenges, online recruitment is often faster, less expensive, and has wider geographic reach than does in-person clinic-based recruitment [29]. Thus, investments in further improving and refining online recruitment strategies may generate high returns for research.

## Supporting information

**S1 File. Online survey instrument.**
(DOCX)

**S2 File. Semi structured interview guide.**
(DOCX)

## Acknowledgments

We would like to thank our colleagues, Sofia Filippa and Samantha Ruggiero, who together conducted five in-depth interviews for this study, as well as Margot Cohen for support in preparing the manuscript.

## Author Contributions

**Conceptualization:** Heidi Moseson, Terri-Ann Thompson, Caitlin Gerdts.

**Data curation:** Carmela Zuniga, Alexandra Wollum.

**Formal analysis:** Heidi Moseson, Jane W. Seymour, Carmela Zuniga, Alexandra Wollum, Anna Katz.

**Funding acquisition:** Heidi Moseson, Terri-Ann Thompson, Caitlin Gerdts.

**Investigation:** Heidi Moseson, Alexandra Wollum.

**Methodology:** Heidi Moseson, Carmela Zuniga, Alexandra Wollum, Terri-Ann Thompson.

**Project administration:** Heidi Moseson, Carmela Zuniga, Alexandra Wollum, Anna Katz.

**Resources:** Caitlin Gerdts.

**Supervision:** Heidi Moseson.

**Writing – original draft:** Heidi Moseson, Jane W. Seymour, Carmela Zuniga, Anna Katz.

**Writing – review & editing:** Heidi Moseson, Jane W. Seymour, Carmela Zuniga, Alexandra Wollum, Anna Katz, Terri-Ann Thompson, Caitlin Gerdts.

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
