## [Decision Letter · Decision Letter 0]

8 Jun 2021

PONE-D-20-38906

“It just seemed like a perfect storm of everything coming together so that I couldn’t have the [abortion]”: a multi-methods study on people who considered but did not obtain abortion care in the United States

PLOS ONE

Dear Dr. Moseson,

Thank you for submitting your manuscript to PLOS ONE. After careful consideration, we feel that it has merit but does not fully meet PLOS ONE’s publication criteria as it currently stands. Therefore, we invite you to submit a revised version of the manuscript that addresses the points raised during the review process.

The manuscript is generally well-written, addressing an important topic and using novel strategies to recruit and collect data from women who considered but did not access abortion services. However, the writing can be improved and additional information provided to make the *Methods* and *Results *sections clearer to the readers. Kindly read the comments of the three independent reviewers and make the recommended revisions.

Please submit your revised manuscript by July 18, 2021. If you .will need more time than this to complete your revisions, please reply to this message or contact the journal office at plosone@plos.org. Please include the following items when submitting your revised manuscript:

We look forward to receiving your revised manuscript.

Kind regards,

Patricia Ongwen, MBChB, MPH

Academic Editor

PLOS ONE

Journal Requirements:

2. We note that consent for the survey was described as written, but that the questionnaire was described as taking place online, therefore please clarify whether consent was digital. Please also state in the Methods:

- Why written consent could not be obtained from interview participants

- Whether the Institutional Review Board (IRB) approved use of oral consent for the interview, and digital consent for the survey

- How oral consent for the interview was documented

For more information, please see our guidelines for human subjects research: https://journals.plos.org/plosone/s/submission-guidelines#loc-human-subjects-research

4. Please include a copy of the interview guide used in the study, in both the original language and English, as Supporting Information, or include a citation if it has been published previously.

6. We note that Figure 2 in your submission contain map images which may be copyrighted. All PLOS content is published under the Creative Commons Attribution License (CC BY 4.0), which means that the manuscript, images, and Supporting Information files will be freely available online, and any third party is permitted to access, download, copy, distribute, and use these materials in any way, even commercially, with proper attribution. For these reasons, we cannot publish previously copyrighted maps or satellite images created using proprietary data, such as Google software (Google Maps, Street View, and Earth). For more information, see our copyright guidelines: http://journals.plos.org/plosone/s/licenses-and-copyright.

6.1.    You may seek permission from the original copyright holder of Figure 2 to publish the content specifically under the CC BY 4.0 license. 

6.2.    If you are unable to obtain permission from the original copyright holder to publish these figures under the CC BY 4.0 license or if the copyright holder’s requirements are incompatible with the CC BY 4.0 license, please either i) remove the figure or ii) supply a replacement figure that complies with the CC BY 4.0 license. Please check copyright information on all replacement figures and update the figure caption with source information. If applicable, please specify in the figure caption text when a figure is similar but not identical to the original image and is therefore for illustrative purposes only.

Reviewers' comments:

Reviewer's Responses to Questions

**Comments to the Author**

1. Is the manuscript technically sound, and do the data support the conclusions?

Reviewer #1: Yes

Reviewer #2: Yes

Reviewer #3: Yes

2. Has the statistical analysis been performed appropriately and rigorously? 

Reviewer #1: Yes

Reviewer #2: N/A

Reviewer #3: Yes

3. Have the authors made all data underlying the findings in their manuscript fully available?

Reviewer #1: Yes

Reviewer #2: No

Reviewer #3: No

4. Is the manuscript presented in an intelligible fashion and written in standard English?

Reviewer #1: No

Reviewer #2: Yes

Reviewer #3: Yes

5. Review Comments to the Author

Reviewer #1: Abstract

No comments

Introduction

- Line 58: period at the end of the sentence. This study that is mentioned is also not cited.

Materials and Methods

- Recruitment

o Line 70: Short recruitment period? Why?

o Line 72: Space between Google and Ads

o Line 74: Why only in English and Spanish?

o Line 79-80: In a supplemental text or appendix? Where has it been described?

- Survey

o Line 84: Inclusion and exclusion criteria is unclear. A flow chart might be helpful. It’s unclear if individuals who have ever had an abortion during the study time frame was included. If subjects who had an abortion were excluded why? It is possible that circumstances changed.

o Line 92: Why could they not go back to revise their answers?

o Line 99-101: If part of the goal of this study was to also look at the feasibility of recruitment via Reddit, Facebook and Google Ads, that should be explicitly stated. Right now, it looks like it is just an excuse you provide to justify why you do not have a target sample size. The study should be explicitly stated as a an exploratory/feasibility study, especially because there

- In-depth interviews

o No comment

- Ethical review

o No comment

Results

- Survey results

o Participant characteristics

Since you have a figure 1, Lines 137-142 can be shortened.

You’ve highlighted the regions where participants are from in Table 1 so figure 2 seems unnecessary to me

o Barriers to abortion care

Presentation of the results are a little confusing. Consider presenting the percentages primarily with n in parentheses

- In-depth interview results

o Participant characteristics

No comments

o Barriers to abortion care

No comments

o Consequences of failure to obtain abortion

No comments

Discussion

I would provide more detail about the feasibility of your recruitment methods since you mention this in your methods section. How does it compare to other studies that recruit in this way? Are you able to recruit a generalizable population? You mention that it the recruitment was successful, but how do you know that? Do you have anything to compare it to? Your sample size is small, is this comparable to other studies that have used these recruitment methods. Elaboration is needed.

Reviewer #2: Overall, this feels like two papers in one: a methods paper about online recruitment methods for this target population and a descriptive/qualitative analysis of barriers experienced by this population. If the intention of this paper is to explicitly report the test of the hypothesis stated on p 2 line 64-67, (minor) revisions are needed to make this more explicit. The current emphasis is on describing the barriers experienced by this population, rather than drawing conclusions about how well these methods identified this understudied population AND/OR whether the (qualitatively or quantitatively) different experience of this population offers critical insights into barriers to care that might be missed by only studying populations captured by more traditional recruitment methods.

It wouldn't take much work to round out either or both of these analyses. For the methods portion, it would be nice to see a little more about the feasibility/lessons learned about these methods, perhaps after providing a little more data (e.g., the relative contribution and value of each recruitment source, measures of reach or efficiency like recruitment time, screen positive rate, missingness, representativeness, etc.) and comparing in some way to other recruitment methods. For the barriers portion, it would be nice to see some analytic conclusions about how the population composition and the barriers experienced by this population compare (in number or type) to other, better characterized populations desiring pregnancy termination. These comparisons could be narrative (rather than statistical) as long as they were well cited and clear.

Additional questions, thoughts, and suggestions are included in comments on the manuscript attached.

Reviewer #3: Overall comments

This is a well-written paper describing a project that recruit people who did not seek abortion care and present at a facility but for whom abortion might have been the best option if not for myriad barriers to accessing care, which is an understudied population. The authors expand on the literature by using novel recruitment strategies to recruit from non-clinical sources. Below I provide detailed feedback on the manuscript, which is mostly minor, but I would like some additional information on the performance of the specific recruitment approaches to strengthen this contribution to the literature to inform related work moving forward.

Specific comments

• Line 58: The authors mention a study that used Google Ads but provided no details or citation. Describe these authors’ experience/findings since that design seems the most relevant to this study.

• Line 72: Was there no Reddit thread for abortion specifically? If there was, why was it not used for recruitment? Would provide this information as readers may question why a more specific “abortion” Reddit thread was not used for recruitment.

• Line 78: Missing the word “to” in “the opportunity complete”.

• Line 98: Was there just one $50 gift card up for raffle? Clarify the incentive for the quantitative component.

• Line 108: Can authors clarify whether all participants who completed the quantitative survey were asked if they were interested in participating in the in-depth interviews? And then if they replied yes, investigators reached out to all of them? Unclear about eligibility/recruitment for this component.

• Line 125: Were quantitative data analyzed before conducting the qualitative interviews? Curious if those findings informed the development of the in-depth interview guide.

• Lines 140-142: From the 217 screened respondents who indicated abortion may have been the best options, how many were further screened out based on age, residence, abortion clinic presentation? Want to know among eligible respondents, what percent the 98 (and 90) represent. Also, in reviewing Figure 1 later in the manuscript, I don’t see the 217 number. Would ensure numbers reference in text reflected in Figure 1. The figure also makes clearer that 143 were actually eligible and 43 didn’t consent, so response rate among eligible respondents was 63% (90/1430?

• Table 1: Do authors have information on when the pregnancy about which they are discussing occurred? Would be particularly useful in interpreting the data on “proportion of time you had enough money to meet basic needs in last month” given that may not reflect the economic situation for many if their pregnancy was not recent.

• Table 1 and associated text: Since authors are using different recruitment strategies to identify an understudied population, would like some information on which of the three approaches were most effective. Authors could add information about recruitment source in Table 1 (Facebook, Google Ads, Reddit) and perhaps also describe broader response rates for each in relation to Figure 1, which just provides overall numbers. This can inform future research, which authors refer to at the end of the discussion but with little details re: which specific strategy worked best.

• Line 170-175: Are these all reasons that authors categorized as “another barrier”? Clarifying where they appear in Table 2 would be helpful.

• Table 3: Missing “0” in N column for “Less than high school graduate”

• Lines 210-212 and 217-219: Authors earlier indicated that all in-depth interview respondents determined their pregnancy before 12 weeks yet these sentences suggest they only had a “few weeks” to decide whether to abort before gestation age limits in the state precluded this option. While authors likely aren’t able to identify participants’ specific states, would be helpful to know what the gestation limits are since these limits seem particularly low if participants only had a few weeks to determine whether to have abortion/how to pay (or maybe I am interpreting “few” to strictly?). Obviously misinformation about gestational limits impacting their decision is a different matter.

• Lines 304-309: What a powerful quote.

• Discussion: Would highlight that although many of the barriers identified are similarly those identified in prior research involving patients recruited from clinics, these barriers may be even greater for the population in this study. Or that the motivation to obtain an abortion is lower. Can’t tell from these findings exactly the role that these two factors played but think it’s worth mentioning in the interpretation of study results.

• Line 355: “(e.g., 28)”? Did authors mean to include text after “e.g.”?

6. PLOS authors have the option to publish the peer review history of their article (what does this mean?). If published, this will include your full peer review and any attached files.

Reviewer #1: No

Reviewer #2: No

Reviewer #3: No

---

## [Author Response · Author response to Decision Letter 0]

11 Nov 2021

Journal Requirements:

Thank you for sharing these guidelines. We have gone through the manuscript and updated section headings and figure captions to match these requirements.

2. We note that consent for the survey was described as written, but that the questionnaire was described as taking place online, therefore please clarify whether consent was digital. Please also state in the Methods:

- Why written consent could not be obtained from interview participants

- Whether the Institutional Review Board (IRB) approved use of oral consent for the interview, and digital consent for the survey

- How oral consent for the interview was documented

For more information, please see our guidelines for human subjects research: https://journals.plos.org/plosone/s/submission-guidelines#loc-human-subjects-research

Thank you for this guidance. We have revised the methods section in lines 147-151 to address these points. It now reads as follows: “To minimize risk of breach of participant confidentiality, all participants gave digital consent to participate in the survey. Similarly, participants gave verbal consent to participate in the interviews; verbal consent was audio-recorded in the audio file, and also documented in a printed consent form, signed and dated by the interviewer. This study, including informed consent processes, was approved by the Allendale Investigational Review Board.”

We are happy to provide this information. We have uploaded the survey and interview guides as supporting information. 

4. Please include a copy of the interview guide used in the study, in both the original language and English, as Supporting Information, or include a citation if it has been published previously.

Per above, we are happy to do this and have uploaded the interview guide as supporting information.

We are unable to share data publicly because of ethical and legal restrictions imposed by the ethics committee that permitted data to be available only to members of the research team named in the institutional review board protocol; and further, because the data contain potentially identifying and sensitive participant information. Interested researchers can request access to the data via addition to our IRB protocol by contacting our data access committee, chaired by Allie Wollum (awollum@ibisreproductivehealth.org). We have added this information to our revised cover letter as well. 

Thank you.

6. We note that Figure 2 in your submission contain map images which may be copyrighted. All PLOS content is published under the Creative Commons Attribution License (CC BY 4.0), which means that the manuscript, images, and Supporting Information files will be freely available online, and any third party is permitted to access, download, copy, distribute, and use these materials in any way, even commercially, with proper attribution. For these reasons, we cannot publish previously copyrighted maps or satellite images created using proprietary data, such as Google software (Google Maps, Street View, and Earth). For more information, see our copyright guidelines:http://journals.plos.org/plosone/s/licenses-and-copyright. 

6.1. You may seek permission from the original copyright holder of Figure 2 to publish the content specifically under the CC BY 4.0 license. 

We recommend that you contact the original copyright holder with the Content Permission Form (http://journals.plos.org/plosone/s/file?id=7c09/content-permission-form.pdf) and the following txt:

 Per comments from the reviewers, we have opted to remove the figure as it does not add essential information to the publication.

6.2. If you are unable to obtain permission from the original copyright holder to publish these figures under the CC BY 4.0 license or if the copyright holder’s requirements are incompatible with the CC BY 4.0 license, please either i) remove the figure or ii) supply a replacement figure that complies with the CC BY 4.0 license. Please check copyright information on all replacement figures and update the figure caption with source information. If applicable, please specify in the figure caption text when a figure is similar but not identical to the original image and is therefore for illustrative purposes only.

 Per above, we have opted to remove the figure as it does not add essential information to the publication.

 

Reviewers' comments:

Reviewer's Responses to Questions

Comments to the Author

1. Is the manuscript technically sound, and do the data support the conclusions?

Reviewer #1: Yes

Reviewer #2: Yes

Reviewer #3: Yes

2. Has the statistical analysis been performed appropriately and rigorously?

Reviewer #1: Yes

Reviewer #2: N/A

Reviewer #3: Yes

3. Have the authors made all data underlying the findings in their manuscript fully available?

Reviewer #1: Yes

Reviewer #2: No

Reviewer #3: No

4. Is the manuscript presented in an intelligible fashion and written in standard English?

Reviewer #1: No

Reviewer #2: Yes

Reviewer #3: Yes

5. Review Comments to the Author

Reviewer #1: Abstract

No comments

Introduction

- Line 58: period at the end of the sentence. This study that is mentioned is also not cited.

Thank you for highlighting this. We have added a period to the end of the sentence, and added two references from the study mentioned, including a core finding. The added text reads: “Another recruited currently pregnant participants searching for information on abortion from Google Ads: at the one-month follow-up, less than half of participants (48%) had had an abortion.1,2”

Materials and Methods

- Recruitment

o Line 70: Short recruitment period? Why?

The one-month recruitment period mirrored a similar study conducted with online recruitment to measure self-managed abortion experiences. We were interested in learning how many people we could recruit in a short period of time. The sentence in lines 83-84 reads: ‘The one-month recruitment period mirrored a recent abortion-related study that recruited using GoogleAds.23”

o Line 72: Space between Google and Ads

We have added the space – thank you. 

o Line 74: Why only in English and Spanish?

We fielded advertisements and bid on keywords in English and Spanish only, primarily because Spanish is the second most widely spoken language in the United States after English, and also based on research team language abilities, and the need to balance study team bandwidth and budget. Adding each additional language was not just a matter of translation – we also had to think about the words used to search in each language, many of which are not direct translations – rather, entirely different phrases in some settings. We have revised the text slightly to make this rationale more clear (lines 86-87): “Advertisements in English and Spanish (the two most widely spoken languages in the United States) read:” 

o Line 79-80: In a supplemental text or appendix? Where has it been described?

Thank you for catching our omission of the appropriate reference here. We have previously described recruitment results in a published manuscript focused exclusively on the performance of each of the three recruitment platforms: Facebook, Google Ads and Reddit. We have revised the text to make this clearer, and added the reference to the publication. The revised text in lines 91-92 reads: “Further details of recruitment methods and results have been previously published.[23]”

- Survey

o Line 84: Inclusion and exclusion criteria is unclear. A flow chart might be helpful. It’s unclear if individuals who have ever had an abortion during the study time frame was included. If subjects who had an abortion were excluded why? It is possible that circumstances changed.

Thank you for highlighting that we could be clearer in this section. We have added a revised flow chart to clarify (Figure 1), and also revised the text. To the reviewer’s question: yes, people who ever had an abortion in the five year study time frame were excluded. The rationale was that we were trying to evaluate whether these online platforms could successfully recruit from a narrowly defined population that had previously been excluded from research. If a person had been to an abortion clinic in the past five years, there is a chance they were recruited into a clinic-based study about abortion access, and at least once had successfully navigated some of the barriers to care. For future research, the reviewer is completely correct that it is important to expand these narrow criteria to include anyone who sought abortion care (whether they obtained it or not) – because yes, circumstances can and do change and we want to capture the fullest breadth possible of experiences. For this particular study, however, because we aimed to test whether we could recruit this previously excluded population, we focused only on those who had not obtained an abortion at all in the past five years. The revised text is in lines 94-98 and reads: “To recruit the narrowly defined population of interest, the survey screener identified eligible participants who were: aged 15 to 49 years old and English or Spanish speaking US residents. In addition, respondents had to report at least one pregnancy in the past five years for which they felt abortion was the best option, but did not obtain an abortion (nor visit an abortion clinic) for any pregnancy in the past five years.”

o Line 92: Why could they not go back to revise their answers?

We programmed the survey in this way to prevent fraudulent submissions, i.e. respondents trying to “game” the survey to become eligible, continually changing their responses until they can get through as eligible and enter for a gift card. We acknowledge that this decision is a trade-off, because some respondents genuinely enter a response in error and wish to go back and fix it. However, with our experience with prior online surveys, this feature helps to minimize fraudulent submissions, and it is our hope that this balances out against the loss of any errors in data some participants might wish to correct. We do offer study team contact information at the end of the survey in case any participant wishes to contact us to correct an error in their survey data. We have added language to explain this in lines 105-106: “To minimize fraudulent responses designed to avoid skip logic patterns, respondents could not go back to revise responses.”

o Line 99-101: If part of the goal of this study was to also look at the feasibility of recruitment via Reddit, Facebook and Google Ads, that should be explicitly stated. Right now, it looks like it is just an excuse you provide to justify why you do not have a target sample size. The study should be explicitly stated as a an exploratory/feasibility study, especially because there

We appreciate this comment from the reviewer. This was indeed a feasibility study, as we address in the discussion, but it was an error to not introduce this earlier in the text. We apologize for the lack of clarity, and have revised throughout the text to make this clearer. For this particular line of text with regard to the sample size, we stated in our IRB protocol and grant proposal that we aimed to recruit at least 10 participants per recruitment method (or a minimum of 30 participants total across Facebook, Google Ads, and Reddit). We initially did not provide this detail in the manuscript, but the reviewer’s point emphasizes the appropriateness of sharing this information. Thus, we have revised the line in question to read: “For this feasibility study, we aimed to recruit a minimum of 10 participants per recruitment platform, or a minimum of 30 total participants across Facebook, Google Ads, and Reddit.” (lines 113-115).

- In-depth interviews

o No comment

- Ethical review

o No comment

Results

- Survey results

o Participant characteristics

Since you have a figure 1, Lines 137-142 can be shortened.

This point is well taken. We have considerably shortened the text in the referenced lines. The revised text in lines 157-158 now reads: “Out of 1,254 eligibility screener submissions, the final analytic sample for these analyses included 66 participants (68.5% of those eligible) from 30 states (Fig 1).”

You’ve highlighted the regions where participants are from in Table 1 so figure 2 seems unnecessary to me

We have removed Figure 2, per the reviewer’s point.

o Barriers to abortion care

Presentation of the results are a little confusing. Consider presenting the percentages primarily with n in parentheses

We have revised the narrative description of these results in line with the reviewer’s suggestion, and present the percentages with n in parentheses, except where n’s are available in a table – then, we just present the percentages to simplify the text.

- In-depth interview results

o Participant characteristics

No comments

o Barriers to abortion care

No comments

o Consequences of failure to obtain abortion

No comments

Discussion

I would provide more detail about the feasibility of your recruitment methods since you mention this in your methods section. How does it compare to other studies that recruit in this way? Are you able to recruit a generalizable population? You mention that it the recruitment was successful, but how do you know that? Do you have anything to compare it to? Your sample size is small, is this comparable to other studies that have used these recruitment methods. Elaboration is needed.

We are grateful to the reviewer for these comments. Per our responses to reviewer #2, we appreciate that our initial framing of the paper did not make it clear that the focus of this study is on the secondary hypothesis of the feasibility study: whether we could collect information on barriers to abortion care for this population, rather than on the primary hypothesis of whether these online platforms could successfully recruit this narrowly defined target population. We have revised the introduction and full text to make this focus on the secondary hypothesis and thus on data related to abortion-seeking experiences the clear focus, rather than the results on recruitment methods as those data have been previously published. We have added text in the abstract, introduction and the discussion to make this clearer. For instance, added text in the abstract (lines 23-25) reads: “We aimed to recruit participants who had considered, but failed to obtain, an abortion using three online platforms, and to evaluate the feasibility of collecting data on their abortion-seeking experiences in a multi-modal online study.” Added text in the intro in lines 71-77: “We hypothesized that: (1) we would successfully find and recruit members of this population using online platforms, and (2) would be able to capture new information on the number and nature of barriers to abortion care experienced by this understudied population. We have previously published results regarding the first hypothesis: we found that these platforms can indeed identify and recruit the target population.[23] This study reports results related to the second hypothesis: the ability to collect data on barriers to abortion care from this previously excluded population.” And finally, we also added text in the discussion (lines 334-337) to clarify this focus, which reads: “In this multi-methods feasibility study, we evaluated whether it was possible to collect data on abortion-seeking experiences from an understudied population recruited via three online platforms. In a one-month recruitment period, we succeeded in identifying and enrolling 66 individuals that met these criteria, and in collecting data on the nature and extent of barriers to abortion care that they had faced.”

Reviewer #2: 

Overall, this feels like two papers in one: a methods paper about online recruitment methods for this target population and a descriptive/qualitative analysis of barriers experienced by this population. If the intention of this paper is to explicitly report the test of the hypothesis stated on p 2 line 64-67, (minor) revisions are needed to make this more explicit. The current emphasis is on describing the barriers experienced by this population, rather than drawing conclusions about how well these methods identified this understudied population AND/OR whether the (qualitatively or quantitatively) different experience of this population offers critical insights into barriers to care that might be missed by only studying populations captured by more traditional recruitment methods.

It wouldn't take much work to round out either or both of these analyses. For the methods portion, it would be nice to see a little more about the feasibility/lessons learned about these methods, perhaps after providing a little more data (e.g., the relative contribution and value of each recruitment source, measures of reach or efficiency like recruitment time, screen positive rate, missingness, representativeness, etc.) and comparing in some way to other recruitment methods. For the barriers portion, it would be nice to see some analytic conclusions about how the population composition and the barriers experienced by this population compare (in number or type) to other, better characterized populations desiring pregnancy termination. These comparisons could be narrative (rather than statistical) as long as they were well cited and clear.

We very much appreciate these comments from the reviewer about this manuscript feeling like two papers in one. Indeed, the larger objective of this feasibility study was twofold: (1) evaluate the ability of three online recruitment methods to identify a previously excluded population, and (2) explore whether this population might offer insights into previously missed barriers to abortion care. We have previously published a manuscript on the results of the recruitment evaluation comparing the three online platforms for recruitment in the Journal of Medical Internet Research (2020). The aim of this current manuscript is to explore the second part of our objectives: whether we could collect data on abortion-seeking experiences from this population that might offer new insights into barriers to abortion care. Thus, we have revised the introduction – especially the framing of our hypothesis as highlighted by the reviewer – as well as the rest of the paper and the discussion in particular to focus in on this secondary objective, and thereby hopefully eliminate some of the confusion. In this vein, we have also added discussion of analytic conclusions about how the population and barriers identified differ from prior studies, to the extent appropriate and warranted given the small sample size. These conclusions are located in the discussion section in several points, notably the second paragraph of the discussion, and the first paragraph of the limitations section. The text reframing the hypothesis and focus of the paper can be found in lines 23-25, 71-77, and 334-337, as quoted above in response to reviewer #1’s comments.

Additional questions, thoughts, and suggestions are included in comments on the manuscript attached.

Comments from manuscript copied here:

Abstract, lines 27-29: Framing the results in this way makes the study population look bigger than it actually was. I understand that part of the goal was to assess these digital recruitment methods, but the title and Discussion portion of the abstract are not focused on the method but instead on the content of the responses to these surveys & interviews. 

We appreciate this feedback. The reviewer is correct that the primary goal of this paper is to present results from the surveys and interviews, not to evaluate the recruitment methods as that was done in a prior publication. Initially, we included the information on initial responses and exclusion due to ineligibility to highlight the relatively high proportion (18.2%) of respondents who reported not having obtained an abortion despite feeling it could have been the best option for their pregnancy. This is an important finding and suggests that many people might not be obtaining wanted abortion care. However, we very much appreciate the reviewer’s point that this might mislead readers that the manuscript will present results from a larger study sample than we analyzed for this particular manuscript. Thus, we have revised the text in the abstract to focus on the participants who completed the full survey, including those who participated in the in-depth interviews. The revised text in the abstract (lines 29-32) now reads: “For the primary outcomes of this analysis, we succeeded in capturing data on abortion-seeking experiences from 66 individuals who were not currently pregnant and reported not having obtained an abortion, nor visited an abortion facility, despite feeling that abortion could have been the best option for a recent pregnancy.”

Lines 53-57: May want to highlight why abortion clinics + prenatal clinics is still insufficient. Who does this miss? I would argue some of the people missed are also populations excluded from this study (e.g., people who miscarry, people who are currently pregnant but have not yet entered into care) as well as people who have no source of care or feel they can't access care without increasing risks (e.g., deportation or other privacy concerns). 

This is an important suggestion. We have added text to the introduction where suggested to make this point. The added text in lines 59-62 reads: “However, even the expansion to prenatal clinics may still miss portions of the target population, including people who miscarry, who have not yet entered into care, or who feel they have no source of care or cannot access care without increasing risks of deportation or other privacy concerns.”

Lines 57-58: What did this study find that indicated the need for your study? Why wasn't it sufficient? Also, this is missing a citation (and a period).

Our study was conducted concurrently to the study cited; thus, both studies were responding to the same need to recruit from a broader population of people considering abortion, not just those who made it to care. In comparison to the cited study, however, our study recruited from a broader range of online sites (not just Google Ads, but also Facebook and Reddit). We have added the citation as suggested by the reviewer, and added more relevant context to the sentence. The revised text now in lines 62-64 reads: “Addressing many of these concerns, a recent study recruited currently pregnant participants searching for information on abortion from Google Ads: at the one-month follow-up, less than half of participants (48%) had had an abortion.[15, 16]”

Lines 64-67: If the intention of this paper is to explicitly report the test of this hypothesis, (minor) revisions are needed to make this more explicit. The current emphasis is on describing the barriers experienced by this population, rather than drawing conclusions about how well these methods identified this understudied population OR whether this population appears to experience qualitatively or quantitatively different barriers than the populations captured by more traditional recruitment methods. 

This is a point well taken. We have revised the presentation of our hypothesis to more clearly delineate the overall hypotheses of the feasibility study, and which specific hypothesis is addressed in this manuscript. The revised text in the introduction in lines 67-77 reads: “With this multi-methods feasibility study, we used three online platforms (Facebook, Google Ads, and Reddit) to recruit a narrow population of people who have been left out of most prior research on abortion access: namely, people who had considered, but not obtained an abortion, nor made it to an abortion clinic, and to assess their experiences with accessing abortion care. We hypothesized that: (1) we would successfully find and recruit members of this population using online platforms, and (2) would be able to capture new information on the number and nature of barriers to abortion care experienced by this understudied population. We have previously published results regarding the first hypothesis: we found that these platforms can indeed identify and recruit the target population.[23] This study reports results related to the second hypothesis: the ability to collect data on barriers to abortion care from this previously excluded population.”

Lines 72: Assuming this is the missing citation above?

Actually – a different study that focused only on self-managed abortion, but we have added the appropriate citation to the text above as recommended. Thank you for flagging.

Lines 74: Given the focus on the recruitment method, more details may be warranted here. How were the Reddit campaigns handled?

The Reddit campaigns were handled by members of the research team reaching out to administrators of individual Reddit threads, asking permission to post, and then posting. Given that the previously published manuscript focuses in detail on recruitment methods and results, we opted not to add more detail here as this has been previously covered. To the reviewer’s point, we added language to refer readers to the prior publication where more details are provided on Reddit and other platform recruitment (lines 91-92). 

Lines 79-80: Where have details on study recruitment been described?

Per above, we have added the citation.

Lines 92-93: any cognitive testing?

We did not conduct cognitive testing prior to launching the survey; however, short-answer survey questions allowed participants to provide context for how they interpreted and responded to key questions. We have uploaded the survey and interview instruments as supplemental materials for this paper. 

Lines 100-101: How was reach measured? And what did you find?

Quantitatively, reach was measured in terms of the number of impressions and clicks on advertisements, reported in our prior publication. Qualitatively, reach was measured by whether we succeeded in recruiting anyone who reported not obtaining an abortion for a pregnancy in the past 5 years, despite feeling that abortion could have been the best option for that pregnancy. In short, it was a binary measure of whether we could identify people in this previously excluded population using each of the three platforms, or not. We report on these outcomes in more detail in our prior publication, and have revised the text in this line per a comment from reviewer #1, to add more detail about target sample size (lines 113-115).

Lines 109-110: Cisgender is called out here, but other characteristics of these interviewers might also be relevant. In particular, it could be imagined age, race, region, fluency in participants preferred language, and positioning about abortion might all relate participants comfort during the interview. 

We agree. We did not know if additional detail would be welcome; but given this comment, have provided more detail on interviewer characteristics that could influence positionality vis a vis the interviewees. The revised text in lines 125-127 reads: “Five cisgender women who identified as Afrolatina, Asian, white, and/or Latinx, fluent in English (and two also in Spanish), resided in California or Massachusetts, and were trained in in-depth interviewing, conducted all 30-60 minute interviews.”

Line 138: If you still have access to these "duplicates", they may be worth looking at more closely. This target population is likely to overlap with those of lower SES which may also coincide with group living situations or and more public internet use (e.g., in libraries). How often were the pattern of responses different enough to likely be a different person?

We agree very much this comment, and it is a point we emphasize on our research teams. Correspondingly, we did not rely on IP address alone to identify duplicates. Rather, we flagged anyone with a duplicate IP address, and then carefully reviewed responses for submissions from the same IP address to determine whether it appeared to be a duplicate, or a unique respondent. 

Line 142-145: These populations may also be worth looking at more closely. The miscarriage group are unlikey to be captured by other recruitment methods so may offer new information, and (maybe more pertinent) not all people clearly differentiate between miscarriage and abortion, particularly abortions "obtained" outside of a formal medical setting. Those that are currently pregnant (or anyone excluded because they did visit a facility and/or obtain an abortion) might be a good comparison group for how the primary target population may be similar or different (above and beyond any difference attributable to the novel recruitment method).

We appreciate these comments. We excluded those who reported a miscarriage because, in the few participants for which that was the case, most had scheduled abortion appointments and had no difficulties in doing so, but only reported not obtaining the abortion because they miscarried. In in-depth interviews, we probed as to whether they had done anything to bring on the “miscarriage” – and in all cases, it truly seemed to be a spontaneous miscarriage, rather than something brought on by use of mifepristone, misoprostol, or other means.

The suggestion to look at currently pregnant people/those excluded due to having visited a facility is an interesting one, and one we will explore in the full study being conducted now. Unfortunately, with just 11 and 13 people in each of these groups, the sample size feels too small to identify or draw meaningful conclusions about similarities or differences.

Lines 153-155: How does this compare to demographic characteristics of those obtaining abortions? Those having babies? The lack of Black and Asian women is noted in the discussion, but maybe worth an explicit comparison?

To the reviewer’s excellent suggestion, we have added information on how the racial composition of our sample compares to abortion patients nationally, as well as the only previously recruited sample of general abortion –seekers (recruited via Google Ads) in the discussion section (lines 384-396). This revised text reads: “This feasibility study has limitations. First, the study population is unlikely to be representative of all individuals who considered but did not obtain abortion care. The narrowly defined analytic sample was small, and included no Asian or Black participants. This is in stark contrast to the racial and ethnic composition of abortion patients nationally, among whom approximately 28% are Black, 25% Hispanic, and 6% Asian or Pacific Islander.[27] For a more direct comparison, a recent study that recruited abortion-seekers using Google Ads had greater success in recruiting Black or African-American as well as Asian participants (28.7% and 2.1% of their sample, respectively) – suggesting gaps in our advertisement campaign, rather than the inability of these online methods to recruit a more racially diverse sample.[15] Indeed, the lack of racial and ethnic diversity in our sample may partially have resulted from the set of advertisements most frequently displayed by Facebook, which excluded the advertisement sets with images of Asian and/or Black individuals. Future studies could better control the diversity of advertisements displayed by creating dedicated campaigns for each advertisement with separate, dedicated funds.”

Table 1: 

- missing census region for 9 people: Can this be identified based on IP address?

Unfortunately, due to IRB protocols – we deleted the IP address after data were collected and cleaned (to protect participant identifying information), and thus cannot go back and recreate Census region. 

- Missing relationship status and education: Was information missing from the same 9 people? If yes, any other notable differences in their pattern of responses? Could this have been a problem with the survey or internet stability or something? 14% missing is of interest to anyone wanting to duplicate these methods so you may want to explain this a little more.

Yes, these sociodemographic data were missing from the same nine people. We have reviewed their responses, and these participants answered all core survey questions but skipped the last section on sociodemographic characteristics which came at the end of the survey. These participants came from all three sites Facebook, Google, and Reddit, some took the survey in English and others in Spanish, participated on a range of dates (from 8/20 to 9/11), and provided detailed write-in responses to short answer questions describing different scenarios. Participants spent a range of time in the survey as well. All to say, these seem to be data from participants who either were interrupted, or decided to skip the sociodemographic questions out of privacy concerns or some other reason. We have added context on this to the first section of the results, in lines 161-163. The text reads: “Nine participants (14%) skipped the final survey section on sociodemographic characteristics: two of these respondents took the survey in Spanish and seven in English.”

Barriers to abortion care, line 161: how does this compare to types and number of barriers experienced by women captured by other recruitment methods?

This is a central point that the reviewer raises. We have added more direct commentary on this in the second and third paragraphs of the discussion section. Many of the barriers identified in this study have been identified in previous studies – however, the extent of overlap, and the nature of some of the barriers identified are novel. Some of this text includes the following (lines 345-356): “Utilizing these underutilized online recruitment approaches, we identified many barriers similar to those identified in prior research involving participants recruited from abortion clinics, such as the inability to pay for the abortion, long distances to the nearest provider, logistical difficulties, legal restrictions on abortion, and abortion stigma.[1-10, 13] Similarly, barriers related to inaccurate information about gestational and other legal limits identified in this feasibility study build on prior research that explores information gaps related to finding and accessing abortion care.[6, 8] The magnitude of the barriers identified in this feasibility study, however, and the extent of overlap across barriers may be of greater magnitude than previously measured: nearly one-third of participants reported experiencing three or more barriers. 

Additionally, one barrier identified by this study – medication abortion preference – has not been widely emphasized in the existing literature. …”

182-183 re participant GA among IDI participants : Is this information available for the survey respondents? This would be another potential characteristic to compare between this and better characterized populations. 

We did not collect data on gestational age at time of pregnancy discovery (or of abortion) among survey participants, unfortunately. We asked as to whether later GA was a factor in not obtaining abortion care, but did not inquire about specific gestational age. 

Also, are more specific details about timing of discovery available for the interviewees? Big functional difference between discovery at 11 weeks and 6 days versus 8 weeks, though they are both <12 weeks. Especially because of the observation about medical abortion, if the data is there, might be worthwhile to look into how many discovered their pregnancy before 10 weeks.

We went back through the transcripts, and for the 10 interviews included for this analysis, participants discovered their pregnancies at: 4, 5, 6, 6, 6, 8, 8, 9, 10, and 12 weeks (mean of 7.4 weeks). So, all to say, about half discovered their pregnancies at 8 weeks or later, meaning they had a short window of time to schedule an appointment, raise funds, and manage the logistics of getting there with additional visits etc for a medication abortion within the 10 week window. To provide this context, we have added this detail to the IDI participant characteristics section in lines 203-205: “Five participants discovered their pregnancy at or before 6 weeks gestation, while the other five discovered their pregnancies at 8, 9, 10, or 12 weeks gestation.”

Age of IDIs at preg, lines 182-183: I think I understand why the lower bound is 19 (most recent versus earliest pregnancy for which abortion was considered), but it's a little confusing when the person who was pregnant at 17 is quoted below.

Yes, the reviewer is correct. In the interviews, we consistently asked about age at most recent pregnancy for which abortion was considered, but not all interviewees provided information on age at earliest pregnancy for which abortion was considered. Thus, we provide the data on most recent pregnancy for consistency, and age at other pregnancies when it was available.

Line 190: Given the emphasis on interaction or overlapping reasons, a proportional venn diagram or similar graphic might be nice? Either here, above, or both. Both would show similarities between the full survey sample and the interview sample.

Per this excellent suggestion, we created a venn diagram in Stata to depict the overlap across the three most frequently reported barriers in the survey. This diagram has been added as figure 2, and depicts overlap across financial, stigma, and provider location barriers. While this added figure does not compare interview barriers to those reported in the survey, it highlights the overlap of barriers participants reported in both data collection modes.

Discussion

Lines 311-312: Some points that could have been clearer in the discussion: 

1. How well did you capture this population? 

2. How does this population compare with other better characterized populations? Any observations here about whether the composition or the barriers are different for those that are delayed vs. those that are prevented?

3. What insights does this population offer into less well characterized barriers or the effect of multiple, intersecting barriers? (THIS POINT IS THE MOST COVERED, but still could be more direct.)

4. Was the information gained worthwhile given the effort/learning curve/limitations of these online methods? 

We appreciate these clarifying questions from the reviewer as to the top-line conclusions that are most helpful to emphasize in the discussion. Per our clarification on the focus of this analysis on our secondary hypothesis – whether we could collect data on (potentially novel) abortion experiences from this narrowly defined target population – we have reframed the first section of the discussion to focus on answering questions 3 and 4 from the reviewer. Our previously published paper on recruitment results addresses or primary hypothesis (CAN we identify and enroll this population? Any members of it?) – which speaks to question #1. Given that this population has not been exhaustively documented or studied, we can’t necessarily definitively comment on how well or how representatively we captured this population, but we can and do address how this sample compares to one prior sample recruited in this way (lines 386-392): “This is in stark contrast to the racial and ethnic composition of abortion patients nationally, among whom approximately 28% are Black, 25% Hispanic, and 6% Asian or Pacific Islander.[27] For a more direct comparison, a recent study that recruited abortion-seekers using Google Ads had greater success in recruiting Black or African-American as well as Asian participants (28.7% and 2.1% of their sample, respectively) – suggesting gaps in our advertisement campaign, rather than the inability of these online methods to recruit a more racially diverse sample.[15]” Further, we’ve revised the discussion text to better show what insights this population offers to barriers to abortion care, and added additional language referenced in responses above and to subsequent reviewer comments to indicate why we feel the information gleaned was worth the effort, and how to overcome some of the limitations via lessons learned in this feasibility study.

Lines 352: What other reasons could there be? And how might any of them be addressed? Giving some thoughts on this latter question will help bolster confidence in these novel methods, unless the intention is to leave the reader with the feeling that (these) online methods are of limited utility.

We are fairly certain the failure resulted from the fact that Facebook did not evenly display our advertisements (which included a wide range of images with people presenting as various races/ethnicities). Instead, Facebook only displayed ~2 of the ads (instead of the 10 we developed) – and the 2 they displayed included fairly white and/or Latinx presenting people, and no images of Black or African American, or Asian, individuals. We have added to the discussion (in lines 394-397) a conclusion that future recruitment efforts should utilize individually funded advertisement campaign with dedicated funds for each advertisement individually which could combat this erasure of certain ads. The added text reads: “Future studies could better control the diversity of advertisements displayed by creating dedicated campaigns for each advertisement with separate, dedicated funds, and prioritizing advertisement posting to groups and pages that center the experiences and interests of people who hold multiple racial and ethnic identities other than “white”.” There is also a long standing, warranted mistrust of sexual and reproductive health research on the part of many communities – however, given that other studies have successfully recruited more racially diverse samples via similar platforms, we attribute this failure to the details of the advertisement display algorithm, rather than the recruitment platform itself. 

Lines 354-355: Similar to above... This is everyone's concern about online methods, so it is definitely important to recognize, but probably equally important to directly address. Again, unless the intention is for the reader to conclude that valid, generalizable data is unikely to be derived from studies using online recruitment methods, you may need to give some information here as to why these methods should still be employed (and under what circumstances) and how to use the findings reported here.

This point is very well taken. We have revised this section of the discussion to provide more context and framing around the value of these online recruitment strategies, see lines 413-420: “Despite the above limitations, this feasibility study succeeded in collecting data from a narrowly defined and understudied population recruited via three online platforms, and the findings presented here can inform future research among larger samples to ensure greater diversity across participant experiences and identities, and to uncover lesser studied barriers to abortion care and the ways in which barriers interact to reinforce each other. Despite some challenges, online recruitment is often faster, less expensive, and has wider geographic reach than does in-person clinic-based recruitment.[29] Thus, investments in further improving and refining online recruitment strategies may generate high returns for research.”

Lines 369-371:What was the metric for success in this context? 

Per a response to a prior comment in the methods section, the metric for success was a more or less binary measure of whether we could find and enroll people who fit the eligibility criteria (did not obtain an abortion for a pregnancy in the past 5 years, even though abortion could have been the best option for that pregnancy) using each of the three platforms. We were not sure if people would willingly / openly disclose that the abortion would have been the best option for a recent pregnancy (a pregnancy that they continued and in most cases, are now parenting a child from). We initially hoped to recruit at least 10 from each platform in a one month period, and we did. Further, we succeeded in capturing data from them about their abortion seeking experiences using these online platforms. To make this more explicit, we have revised the text here in lines 334-339 to read: “In this multi-methods feasibility study, we evaluated whether it was possible to collect data on abortion-seeking experiences from an understudied population recruited via three online platforms. In a one-month recruitment period, we succeeded in identifying and enrolling 66 individuals that met these criteria, and in collecting data on the nature and extent of barriers to abortion care that they had faced. Specifically, participants described multiple, intersecting barriers to abortion care which ultimately deterred them from seeking, or prevented them from obtaining, an abortion in the United States.”

Figure 1 flow chart:

1048 excluded due to age, no preg, etc: Would be great to know the breakdown here, i.e., how many were excluded for each reason?

We have revised Figure 1 to provide the detailed information on how many were excluded for each reason.

Screener submission: Would be great to know how many of these came from Google vs Facebook vs Reddit, and whether or not one source was better than another at capturing finding truly eligible people or produced a more diverse/representative final sample.

We have revised Figure 1 to indicate how many came from each platform for the initial screener submission. Beyond screener submissions, we do not break this information down further as it has already been published in our prior publication. 

Reviewer #3: 

Overall comments

This is a well-written paper describing a project that recruit people who did not seek abortion care and present at a facility but for whom abortion might have been the best option if not for myriad barriers to accessing care, which is an understudied population. The authors expand on the literature by using novel recruitment strategies to recruit from non-clinical sources. Below I provide detailed feedback on the manuscript, which is mostly minor, but I would like some additional information on the performance of the specific recruitment approaches to strengthen this contribution to the literature to inform related work moving forward.

Specific comments

• Line 58: The authors mention a study that used Google Ads but provided no details or citation. Describe these authors’ experience/findings since that design seems the most relevant to this study.

We thank the reviewer for catching our omission, and we have provided the citation and additional detail in the introduction (lines 64) and discussion (lines 392), per responses to above reviewer comments as well.

• Line 72: Was there no Reddit thread for abortion specifically? If there was, why was it not used for recruitment? Would provide this information as readers may question why a more specific “abortion” Reddit thread was not used for recruitment.

To the reviewer’s question: yes, there is a Reddit thread for abortion specifically, but it does not allow posts from researchers (or did not at the time). Further, as we wanted to capture people who failed to obtain an abortion, the threads that we selected were broader and likely to encompass people who considered but did not obtain abortion. We have added this in lines 81-83: “Between August 15 and September 15, 2018, we recruited for a brief online survey through advertisements on Facebook, Google Ads, and two Reddit threads (birth control and menstruation; the abortion thread did not allow researchers to post for study recruitment).”

• Line 78: Missing the word “to” in “the opportunity complete”.

We have added in the missing “to”; we thank the reviewer for catching this.

• Line 98: Was there just one $50 gift card up for raffle? Clarify the incentive for the quantitative component.

Yes, due to limited budget for this feasibility study, we offered one $50 gift card for raffle. We have clarified this in line 112-113: “Participants who completed the survey were entered into a raffle for a single $50 gift card.”

• Line 108: Can authors clarify whether all participants who completed the quantitative survey were asked if they were interested in participating in the in-depth interviews? And then if they replied yes, investigators reached out to all of them? Unclear about eligibility/recruitment for this component.

Yes, all participants who completed the quantitative survey were invited to participate in an interview, and anyone who expressed interest was contacted to set up an interview. Not all participants who expressed interest responded to our outreach. We have clarified this in the text in lines 122-125: “All participants who completed the quantitative survey were asked if they were interested in participating in an in-depth interview; if interested, participants provided their name or pseudonym, email address or phone number, and preferred language. The research team contacted all participants who expressed interest to schedule an interview.”

• Line 125: Were quantitative data analyzed before conducting the qualitative interviews? Curious if those findings informed the development of the in-depth interview guide.

The in-depth interview guide was designed at the same time as the quantitative survey instrument. Quantitative data, however, were analyzed prior to qualitative interview transcripts. We have clarified this in lines 142-143: “The survey and in-depth interview guide were designed concurrently to complement one another. We analyzed survey data prior to interview data.”

• Lines 140-142: From the 217 screened respondents who indicated abortion may have been the best options, how many were further screened out based on age, residence, abortion clinic presentation? Want to know among eligible respondents, what percent the 98 (and 90) represent. Also, in reviewing Figure 1 later in the manuscript, I don’t see the 217 number. Would ensure numbers reference in text reflected in Figure 1. The figure also makes clearer that 143 were actually eligible and 43 didn’t consent, so response rate among eligible respondents was 63% (90/143)?

We appreciate these comments on eligibility and where in the process participants were screened out. Our previously published publication contains some of this information, but we have also revised Figure 1 extensively to provide this requested detail. The revised figure is uploaded with this submission, and includes information on how many respondents came from each platform, as well as the numbers excluded for each particular eligibility criteria. To the reviewer’s specific question: the 217 number does not appear in the flow chart because people were additionally screened out after this question before getting to the final eligible number of respondents. Per the comment from the reviewer, we have also included the response rate (98/143=68.5% among eligible respondents) in line 158: “Out of 1,254 eligibility screener submissions, the final analytic sample for these analyses included 66 participants (68.5% of those eligible)…”

• Table 1: Do authors have information on when the pregnancy about which they are discussing occurred? Would be particularly useful in interpreting the data on “proportion of time you had enough money to meet basic needs in last month” given that may not reflect the economic situation for many if their pregnancy was not recent.

Unfortunately, we did not collect data on the specific year in which the pregnancy occurred – only that it was in the past five years. Thus, data about current financial situation may or may not have applied at the time of the pregnancy. That data is included in Table 1 to present a picture of the current profile of study participants.

• Table 1 and associated text: Since authors are using different recruitment strategies to identify an understudied population, would like some information on which of the three approaches were most effective. Authors could add information about recruitment source in Table 1 (Facebook, Google Ads, Reddit) and perhaps also describe broader response rates for each in relation to Figure 1, which just provides overall numbers. This can inform future research, which authors refer to at the end of the discussion but with little details re: which specific strategy worked best.

We very much appreciate this comment. We have added this initial information in the revised Figure 1, and also to Table 1 to provide information on how many were recruited from each platform, as well as how many among the final, eligible, included sample came from each platform. 

• Line 170-175: Are these all reasons that authors categorized as “another barrier”? Clarifying where they appear in Table 2 would be helpful.

Yes, these reasons were categorized as “another barrier”. We have added a note in the text to make this clearer. The revised text in lines 182-183 reads “In open-ended responses provided for the “another barrier” item, 19.7% (n=13) of participants described additional reasons they did not obtain an abortion.” One participant of the 14 who marked this choice did not provide a write-in response; hence, 13 responses.

• Table 3: Missing “0” in N column for “Less than high school graduate”

We have added the 0 where indicated.

• Lines 210-212 and 217-219: Authors earlier indicated that all in-depth interview respondents determined their pregnancy before 12 weeks yet these sentences suggest they only had a “few weeks” to decide whether to abort before gestation age limits in the state precluded this option. While authors likely aren’t able to identify participants’ specific states, would be helpful to know what the gestation limits are since these limits seem particularly low if participants only had a few weeks to determine whether to have abortion/how to pay (or maybe I am interpreting “few” to strictly?). Obviously misinformation about gestational limits impacting their decision is a different matter.

Thanks for this important reflection and question. A few factors were at play in participants feeling that they only had “a few weeks” to gather funds for their abortion, two most frequently:

(1) Some participants found out about their pregnancies later than average, meaning that they truly only did have a few weeks until they reached the legal gestational limit in their state (on the lower end, 14 weeks in states like Indiana, and 20-24 weeks elsewhere). 

(2) Some participants did not have accurate information on the gestational limit in their state, so BELIEVED that they only had a few weeks, when in reality, they would have had longer. However, because they thought they only had a certain window, they gave up pursuing abortion care feeling that they could not raise the needed funds in time.

Rather than naming individual states where participants came from, we’ve instead added some language to clarify how the two named factors above influenced people sense of time pressure in their abortion consideration. The revised text is in lines 233-235 and reads: “Several described that financial challenges were exacerbated by their understanding of the gestational limit on abortion in their state, meaning that they had a short window of time to gather the needed funds, and restrictions on public funding for abortion.”

• Lines 304-309: What a powerful quote.

Agreed. 

• Discussion: Would highlight that although many of the barriers identified are similarly those identified in prior research involving patients recruited from clinics, these barriers may be even greater for the population in this study. Or that the motivation to obtain an abortion is lower. Can’t tell from these findings exactly the role that these two factors played but think it’s worth mentioning in the interpretation of study results.

The reviewer makes an important and relevant point. We have added this context to the discussion in lines 353-354: “However, due to the nature of study questions, it is not possible to disentangle greater or lesser motivation to obtain abortion care from the magnitude of barriers faced.”

• Line 355: “(e.g., 28)”? Did authors mean to include text after “e.g.”?

Apologies. We utilized the “e.g.” here to indicate that the reference we cite is one example of many references that could be cited – it is not an exhaustive list of references. However, as this may confuse readers, we have removed the “e.g.”

---

## [Decision Letter · Decision Letter 1]

17 Feb 2022

“It just seemed like a perfect storm”: A multi-methods feasibility study on the use of Facebook, Google Ads, and Reddit to collect data on abortion-seeking experiences from people who considered but did not obtain abortion care in the United States

PONE-D-20-38906R1

Dear Dr. Moseson,

We’re pleased to inform you that your manuscript has been judged scientifically suitable for publication and will be formally accepted for publication once it meets all outstanding technical requirements.

Kind regards,

Godwin Otuodichinma Akaba, MBBS,MSc,MPH,FWACS

Academic Editor

PLOS ONE

Additional Editor Comments (optional):

The recommended revisons have been extensively implemented.The work is now acceptable for publication.However author should please correct the statement on page 15 line 354-356:Furthermore, given that Internet users are not representative of the generalpopulation (e.g., 28 ), additional work is necessary to recruit for a study that could be generalized to the

larger population of US abortion seekers.

Comment:The e.g in bracket should be deleted

Reviewers' comments:

Reviewer's Responses to Questions

**Comments to the Author**

1. If the authors have adequately addressed your comments raised in a previous round of review and you feel that this manuscript is now acceptable for publication, you may indicate that here to bypass the “Comments to the Author” section, enter your conflict of interest statement in the “Confidential to Editor” section, and submit your "Accept" recommendation.

Reviewer #2: All comments have been addressed

Reviewer #3: All comments have been addressed

2. Is the manuscript technically sound, and do the data support the conclusions?

Reviewer #2: Yes

Reviewer #3: Yes

3. Has the statistical analysis been performed appropriately and rigorously? 

Reviewer #2: Yes

Reviewer #3: Yes

4. Have the authors made all data underlying the findings in their manuscript fully available?

Reviewer #2: No

Reviewer #3: No

5. Is the manuscript presented in an intelligible fashion and written in standard English?

Reviewer #2: Yes

Reviewer #3: Yes

6. Review Comments to the Author

Reviewer #2: As per the authors, data cannot be made fully publicly available in accordance with the terms of their IRB approval.

Reviewer #3: The authors have thoroughly responded to my comments, and have similarly addressed other reviewer concerns. I believe it is acceptable for publication.

7. PLOS authors have the option to publish the peer review history of their article (what does this mean?). If published, this will include your full peer review and any attached files.

Reviewer #2: No

Reviewer #3: No

---

## [Editor Report · Acceptance letter]

24 Feb 2022

PONE-D-20-38906R1 

“It just seemed like a perfect storm”: A multi-methods feasibility study on the use of Facebook, Google Ads, and Reddit to collect data on abortion-seeking experiences from people who considered but did not obtain abortion care in the United States 

Dear Dr. Moseson:

I'm pleased to inform you that your manuscript has been deemed suitable for publication in PLOS ONE. Congratulations! Your manuscript is now with our production department. 

Kind regards, 

on behalf of

Dr. Godwin Otuodichinma Akaba 

Academic Editor

PLOS ONE